# Coping with the COVID-19 Pandemic: Perceived Changes in Psychological Vulnerability, Resilience and Social Cohesion before, during and after Lockdown

**DOI:** 10.3390/ijerph19063290

**Published:** 2022-03-10

**Authors:** Sarita Silveira, Martin Hecht, Hannah Matthaeus, Mazda Adli, Manuel C. Voelkle, Tania Singer

**Affiliations:** 1Sarita Silveira, Social Neuroscience Lab, Max Planck Society, Bertha-Benz-Strasse 3, 10557 Berlin, Germany; hannah.matthaeus@social.mpg.de (H.M.); singer@social.mpg.de (T.S.); 2Hector Research Institute of Education Sciences and Psychology, University of Tübingen, 72074 Tübingen, Germany; martin.hecht@uni-tuebingen.de; 3Department of Psychiatry and Psychotherapy, CCM, Charité–Universitätsmedizin Berlin, 10117 Berlin, Germany; mazda.adli@charite.de; 4Fliedner Klinik Berlin, Center for Psychiatry, Psychotherapy and Psychosomatic Medicine, 10117 Berlin, Germany; 5Department of Psychology, Humboldt-Universität zu Berlin, 12489 Berlin, Germany; manuel.voelkle@hu-berlin.de

**Keywords:** vulnerability, resilience, social cohesion, adaptive coping, mental health, COVID-19 pandemic

## Abstract

The COVID-19 pandemic and associated lockdowns have posed unique and severe challenges to our global society. To gain an integrative understanding of pervasive social and mental health impacts in 3522 Berlin residents aged 18 to 65, we systematically investigated the structural and temporal relationship between a variety of psychological indicators of vulnerability, resilience and social cohesion before, during and after the first lockdown in Germany using a retrospective longitudinal study design. Factor analyses revealed that (a) vulnerability and resilience indicators converged on one general bipolar factor, (b) residual variance of resilience indicators formed a distinct factor of adaptive coping capacities and (c) social cohesion could be reliably measured with a hierarchical model including four first-order dimensions of trust, a sense of belonging, social interactions and social engagement, and one second-order social cohesion factor. In the second step, latent change score models revealed that overall psychological vulnerability increased during the first lockdown and decreased again during re-opening, although not to baseline levels. Levels of social cohesion, in contrast, first decreased and then increased again during re-opening. Furthermore, participants who increased in vulnerability simultaneously decreased in social cohesion and adaptive coping during lockdown. While higher pre-lockdown levels of social cohesion predicted a stronger lockdown effect on mental health, individuals with higher social cohesion during the lockdown and positive change in coping abilities and social cohesion during re-opening showed better mental health recovery, highlighting the important role of social capacities in both amplifying but also overcoming the multiple challenges of this collective crisis.

## 1. Introduction

The coronavirus disease (COVID-19) outbreak and its numerous adverse biopsychosocial consequences is considered an unprecedented challenge for individuals and societies at a global scale [1,2]. As such, the COVID-19 pandemic was proclaimed as a global public health crisis by the World Health Organization (WHO) on 30 January 2020. Several studies already provide seminal evidence of a plethora of direct and indirect pervasive impacts of the COVID-19 outbreak and pandemic-related restrictions. Patterns of increased psychological distress, mental health sequelae and maladaptive behaviors have consistently been reported across different countries [3,4,5], and psychological challenges have particularly been ascribed to the pandemic-related lockdown [6].

To this effect, the social isolation required in lockdowns has added to the mental health challenges by affecting the very fabric of our social lives [7,8,9]. Therefore, it is of equal importance to discover both factors and processes of resilience that allow one to buffer the detrimental effects of such collective stressors on mental health, as well as social cohesion factors, which refer to the experience of social integration and stability. Although several studies have focused on psychological aspects of trait and state vulnerability [3,4,5] or resilience during the pandemic [10,11,12], social cohesion changes [13] and even the relationship of resilience and social cohesion on a community level [14], no study so far has integrated all these crucial aspects including indicators of psychological well-being and vulnerability, resilience and facets of social cohesion in one single study based on a longitudinal design focusing on their structural interrelationship over the duration of different phases of the COVID-19 pandemic. To fill this gap, the present CovSocial project investigated and integrated a broad range of psychological indicators of vulnerability, resilience and social cohesion to determine their interactions and changes before, during and after the first COVID-19-related lockdown in Berlin, Germany.

Such a multidisciplinary and integrative approach aims at determining which factors of resilience and/or social cohesion may protect people from or put people at risk of suffering from mental health challenges when exposed to pandemic-related measures, such as a lockdown associated with social isolation and other restrictions.

Whereas vulnerability and resilience research has a long tradition in psychological science [15,16,17,18], and has acknowledged the role of certain social behaviors as psychobiological health-promoting stress responses [19], the academic discourse on the construct of social cohesion, even though partly rooted in social psychology, has lately been rather concentrated in the social, political and economic sciences [20]. However, many of the public health measures during the COVID-19 pandemic caused severe disruptions in peoples’ social lives and may thus have affected parameters of social cohesion, which in turn may have direct implications for psychological vulnerability and mental health. Generally, social cohesion refers to multidimensional and multilevel mechanisms that create the experience of social unity [21]. Social cohesion may on the one hand serve as a protective factor for communities in times of collective stressors [22,23,24,25], yet on the other hand is likely to be compromised by the social impacts of the pandemic [13]. To better understand the influence of collective stressors such as pandemic-related lockdowns on the well-being and the social lives of people, the current study embarked on integrating notions of social cohesion in existing psychological frameworks of vulnerability and resilience. More specifically, to allow for an integrated psychological framework, the CovSocial project focused primarily on the assessment of subjectively relevant dimensions of social cohesion such as feelings of belonging, trust and social engagement and interaction; that is, dimensions that could be assessed with subjective self-report trait measures as well as psychological state questionnaires during the pandemic.

The main study goals were to identify a broad spectrum of state measures of relevance to psychological vulnerability, resilience and social cohesion, and to systematically investigate the structural and temporal relationship between these concepts in the context of the first pandemic-related lockdown in Germany; specifically, before the pandemic in January 2020 (T1), during the first lockdown from mid-March 2020 to mid-April 2020 (T2) and after re-opening in June 2020 (T3) in a heterogeneous Berlin population ranging from 18 to 65 years of age.

Due to the novelty of our approach and the variety of indicators used, we first aimed at identifying and validating the best factor structure for our indicators reflecting our three main constructs. Second, based on these findings, we investigated how vulnerability, resilience and social cohesion change over the three measurement time points during the pandemic: before, during and after the first lockdown in 2020. Third, using an individual difference approach, we explored the interrelationship of the three constructs over time, aiming to determine whether increases in resilience and social cohesion could help buffer the detrimental effects of this collective stressor on mental health and psychological vulnerability, or, in contrast, whether these were equally affected by the lockdown, since it is associated with social isolation and other social restrictions.

### 1.1. Psychological Vulnerability

The concept of vulnerability refers to the susceptibility to an undesirable outcome [26]. In this sense, vulnerability is reflected in biological, emotional, cognitive or behavioral stress responses that do not lead to successful adaptation or self-regulation and ultimately contribute to the development of disorders and diseases [15]. While particularly uncontrollable or chronic stress and environmental adversity challenge processes that are relevant for adaptation by interrupting habitual functioning, psychological approaches to vulnerability highlight the importance of subjective stress perception and evaluation [27,28].

In line with this, impacts of the COVID-19 pandemic have been found to translate into a range of maladaptive emotional and behavioral stress responses that can serve as indicators of psychological vulnerability. These include impacted emotional and cognitive functioning [29], increased psychopathology [3,4], addictive behaviors including alcohol and substance abuse as well as internet addiction [30,31,32], aggression and domestic violence [33,34], or noncompliance with public health measures [35]. In German samples, an increase in psychopathology during the pandemic was found in approximately 10% of study participants [36,37]. It has further been suggested that impacts of the COVID-19 pandemic on mental health and well-being are particularly related to fears, uncertainties and perceived loneliness during social distancing and isolation [38,39,40] and to news consumption [41,42]. Thus, it seems that pandemic-related collective stressors such as lockdowns have affected a wide range of factors related to mental health challenges.

Therefore, and in contrast to vulnerability research focusing only on symptomatology of mental health disorders [15,18], we opted for a broad conceptualization of vulnerability including subclinical levels of depression and anxiety; alcohol, substance and internet addiction; stress-related somatic problems; subjectively perceived stress; experiences of aggression, loneliness, fears and COVID-19-specific burdens; as well as excessive news consumption and noncompliance with public health measures to derive representative and quantitative continuous measures of vulnerability based on indicators that are also relevant to the special circumstances of the COVID-19 pandemic. Such an approach is in line with other theoretical frameworks promoting general stress-related processes that are relevant to transdiagnostic dysfunctions [43]. Accordingly, one goal of this study was to test the validity of the broader conceptualization of psychological vulnerability assuming that all these indicators may even be reflected by a single latent vulnerability factor, which in turn should then also show structural stability over the three different time points of the present study.

### 1.2. Resilience

In psychological science, the term resilience describes either a set of individual characteristics that enable individuals to successfully maintain or recover well-being and mental health in the face of adversities [44], or the process of recovery after stressor exposure itself, which is defined by outcome-related states over the course of time [45]. As such, it can be viewed as trajectories of either stable mental health over time or rapid recovery after a stressor [46,47]. This idea of dynamic outcome trajectories is also mirrored in views on resilience as a capacity that develops over time, and implies that it can be modified and improved [47,48,49]. Since the defined outcome focuses on an absence as compared to the existence of mental health symptoms in the aftermath of severe stress, resilience has thus been empirically linked to vulnerability in a dichotomous way [50].

Since both psychological vulnerability and resilience refer to dynamic responses to stress and adversities, several integrative frameworks of those two constructs exist to date. Vulnerability and resilience have partly been conceptualized as complementary counterparts, in that higher levels of resilience are indicative of less vulnerability and vice versa [51]. Despite the commonalities, however, empirical evidence highlights that resilience cannot merely be defined as an absence of vulnerability, since it includes processes that uniquely relate to active coping, adaptation, stress recovery and the prediction of mental health [52,53,54].

In line with this, resilience has been described in conjunction with adaptive capacities, the skills and abilities necessary to learn and adjust to changes, which have been ascribed a key role in transition and resilient recovery [55,56,57]. The health-focused paradigm shift of resilience research gave rise to the notion of transdiagnostic adaptive mechanisms that enable mental well-being in response to stress and adversities [58,59]. Drawing from appraisal theory, any stress response is crucially influenced by a subjective, context-dependent and multidimensional evaluation of the situation, which is dynamic and integrative by nature [60]. In line with this, individual tendencies to evaluate stressful situations as positive have repeatedly been proposed to be of crucial importance in promoting resilient trajectories [43,61]. These psychological strategies have also been described as active processes that shape an individual’s stress response in ways that foster adaptation and resilience [62]. A cognitive focus on positive aspects may culminate in the transformation of adversity into a source of positive change and individual growth [63].

Such frameworks of stress resilience thus provide a rationale for the role of appraisals and beliefs that available resources can be accessed to defy uncertainties and a loss of control faced by the COVID-19 pandemic [64]. Indeed, self-efficacy beliefs and positive appraisal style have been found to promote resilience in cross-sectional COVID-19 studies [65,66]. Other cross-sectional studies in the context of the pandemic could show inverse relationships between mental health and optimism [67] or life satisfaction [68].

In the current study, we used a multi-faceted approach to resilience, assessing it (a) as a trait, (b) as a dynamic state-variable that can change over time and (c) as adaptive coping strategies. Trait resilience was measured through psychological trait measures of a general ability to recover from stress, optimism, satisfaction with life and self-compassion and is part of another paper [69]. State-related resilience measures included speed and ease of stress recovery, optimism, life satisfaction, self-efficacy beliefs and perceiving crises as a chance. Adaptive coping capacities that were found to form specific abilities on the trait level were assessed on the state level as well, and were considered to be conceptually related to cognitive resilience-promoting strategies such as perceiving crises as a chance, optimism and self-efficacy beliefs.

Another goal was to determine whether there is evidence for one single resilience factor on the state level or two different factors, similar to our findings on the trait level [69], where one factor reflects adaptive coping capacities and the other reflects resilient person characteristics as opposite to general vulnerability; that is, resilience and vulnerability trait measures loading on one single resilience-vulnerability factor.

### 1.3. Social Cohesion

The term social cohesion was coined by the sociologist Émile Durkheim [70] to describe the interdependence between members of a group marked by mutual support and shared resources. In the field of psychology, social cohesion was first described in terms of shared experiences and thoughts [71] or shared characteristics and thus identification with other group members [72]. From an outcome-related perspective, social cohesion was proposed to result from each members’ attitude towards and motivation to affiliate with a group [73]. Even though a clear conceptualization is still lacking, the contemporary scientific discourse defines social cohesion as an indicator of togetherness in a society, and as such, it revolves around levels of interaction and integration, civic engagement and identity [20,74]. Social cohesion is a multidimensional and multi-system construct that comprises processes in micro (e.g., families, relationships), meso (e.g., communities, neighborhoods, institutions) and macro (e.g., nations) systems of a society, which can be measured subjectively; that is, in peoples’ attitudes and beliefs, or in objective behavioral manifestations thereof [25,75]. Social cohesion has further been described on levels of individuals, communities or institutions [75], as well as on a horizontal dimension, which reflects the cohesion between members of a society, and a vertical dimension, which describes cohesion between the citizens and a state [20]. As we aim at integrating this broad and heterogeneous construct into the recent psychological literature and relate it to psychological constructs of resilience and vulnerability as measured through self-report questionnaires, the current operationalization of social cohesion focuses on those subjectively experienced attitudes and behaviors that reflect central psychological dimensions of social cohesion that can be assessed through self-report questionnaires as opposed to objective markers of observed behaviors or economic dimensions such as economic or societal inequality indicators.

One of the reoccurring dimensions in the social cohesion literature is a sense of belonging [20,75], which has also been referred to as identity [20] or identification [74]. A sense of belonging has been described as the feeling of inclusion into the system or environment in which an individual is operating [76]. The degree of belonging also affects subjective perceptions of social resources and support that are available in times of need [21,76].

Trust has been proposed as another crucial aspect of social cohesion, including both interpersonal and institutional trust [20,21,77]. The dimension of trust relates to how predictable other peoples’ behavior and their underlying good intentions are perceived as by an individual [78]. Trust between individuals in a society is also based on implicit beliefs of shared norms and values [79].

Another dimension concerns social interactions, also called social relations [74], which can be seen as driving forces to remain in a social group [25]. Social interactions concern both the vertical and horizontal dimensions of social cohesion [20]. Other terms used in the context of social interactions are social networks, subsuming quality and quantity of social interactions, or social capital, a term that does not only refer to the connections themselves, but also to values and norms that stem from them [80]. Again, the construct of social cohesion stresses not only social interactions within a social group (i.e., bonding social capital), but also those between different groups (i.e., bridging social capital), addressing levels of inclusion and tolerance [81].

Lastly, we identified social engagement consisting of social and political participation [20,75] and prosocial behavior or the act for the needs and benefits of the group [74] as a fourth crucial aspect of social cohesion. This indicator refers to behavioral manifestations of prosocial motivations, also described as the willingness to participate and help [20] or orientation towards the common good [74].

Certainly, frameworks of social cohesion also include other objective dimensions such as network size, degree of (in)equality in terms of opportunities and a distribution of resources such as education, employment, income or health care [20,74,82]. Given the focus on subject psychological dimensions of social cohesion, we derived four key dimensions from major contemporary conceptualizations: (1) belonging, (2) trust, (3) social interactions and (4) social engagement.

Another goal was to search for validation of the present novel broad operationalization of social cohesion as a psychological construct based on those four subjective psychological core dimensions. A further open question was whether—in analogy with previous psychometric research in cognitive intelligence [83]—these four dimensions of social cohesion (belonging, trust, social interaction and social engagement) could, on a higher level, be further summarized under a general social cohesion factor or rather form distinct, unique, fully independent factors.

### 1.4. Psychological Vulnerability, Resilience and Social Cohesion in the Context of the COVID-19 Pandemic

In line with the existing evidence of impacted mental health [3,4] and other maladaptive stress responses capturing a broad variety of vulnerability indicators [30,31,32,33,34], we expected psychological vulnerability to increase on a latent level during the first pandemic-related lockdown. Note, however, that recent longitudinal studies on mental health impacts of the pandemic in German samples found no overall decrease in mental health status [37] or even an increase from pre- to post-lockdown [36], with only 10% of participants showing mental health issues during the lockdown [36,37]. Given that we focused on a wide range of indicators that included both sub-clinical measures of mental health challenges as well as COVID-19-relevant stress- and burden-related markers, and not just on psychopathology, we assumed that on average, psychological well-being should suffer in the pandemic period.

For psychological resilience, on the other hand, we expected that resilience processes are reversely associated with trajectories of mental health challenges, and can predict better stress recovery. Indeed, it was found that a lockdown-related increase in psychopathology from February to March 2020 was negatively correlated with an individual’s belief that life challenges reflect a potential source of growth [37]. However, while resilience was partly found to overall decrease during the lockdown [11], adaptive abilities were found to be relatively stable over the course of the lockdown in Germany [37]. It therefore remains an open question whether aspects of resilience changed in terms of mutual interdependence and in terms of how far these processes may contribute to or buffer against detrimental impacts of the pandemic on psychological vulnerability.

Due to the repercussions of the COVID-19 pandemic on peoples’ social lives, social cohesion is expected to be impacted by the pandemic and pandemic-related lockdown. While overall concerns have been raised that societies and communities may be prone to fall apart due to the pandemic [13], other theoretical frameworks and empirical evidence from disaster research suggest an increase in social cohesion as a stress response. For example, the “tend-and-befriend” hypothesis claims that prosocial behavior increases as a response to stress, and functions as an evolutionary developed protective pattern of affiliation and care [84,85]. Stress exposure has also been found to strengthen trust, trustworthiness and cooperation [86]. Additionally, post-disaster research shows that experiences of collective stressors can led to an increase in social connections and prosocial behavior [87]. Due to the novelty of the COVID-19 pandemic, particularly with regard to the concomitant restrictions imposed on peoples’ social lives, there remains a gap in knowledge in which ways social cohesion was affected and limited by the pandemic-related lockdown.

Another important question referred to the interaction between vulnerability, resilience and social cohesion during the lockdown. Vulnerability frameworks have integrated social factors such as isolation and loneliness as potential risk factors for developmental trajectories of poor mental health to the point of mortality [88]. In the context of COVID-19, sociodemographic and psychosocial determinants of physical and mental health such as social support, social network quality, socioeconomic status, stigmatization, ethnicity or unemployment have become a realm for the definition of risk groups [89,90]. Vulnerability can thus be expected to increase during the pandemic-related lockdown in direct relation to curbed social interactions, social distancing and isolation, aggravated economic inequalities, segregation and racism [91], and economic hardship amplified by job loss [92]. This would suggest that vulnerability and social cohesion show reverse time-courses, in that vulnerability increases while social cohesion decreases during the lockdown. On the other hand, social cohesion has also been conceptualized as a protective factor against detrimental effects of stress [85,86]. In that sense, high levels of social cohesion could buffer against increases in vulnerability levels due to the pandemic.

From the perspective of the resilience research, particularly social support and attachment have long been proclaimed as a protective factors in times of stress [48,93]. Besides those resources, the experience of social justice and sense of belonging are also proposed to lead to resilient outcomes [94]. Being embedded in a social network is considered a powerful resource for sustained health and well-being [95]. In support of that, it has been shown that greater social connectedness was associated with lower levels of psychological distress during the COVID-19 pandemic-related lockdown [96], and studies on environmental disasters provide evidence that social network support can have a bolstering effect on such a stress response [97,98,99]. In line with this, resilience processes in dealing with collective stressors may be fostered by the availability of social resources [100]. Interestingly, many examples of disaster research show that collective stressors can actually strengthen social cohesion and thereby promote resilience [97,98]. Particularly, prosocial behavior and social engagement were found to increase in times of crises, fostering social support networks and a sense of belonging [101,102,103]. However, the pandemic, in contrast to natural disasters, came with governmental restrictions on our ability to socially interact in person and thus with potential limitations on our ability to engage in prosocial collective behaviors. Indeed, studies show that psychological resilience during the COVID-19 pandemic was adversely affected by dynamic social factors and restrictions [11].

The direction of change in social cohesion during the pandemic and its contribution to changes in vulnerability and resilience processes remain unknown. Thus, one of our main goals was to shed light on how mental health changes towards dysfunction and improvement before, during and after the lockdown are related to changes in social cohesion.

In conclusion, the present study pursued several goals. In the first step, we aimed at identifying and validating the most probable factor structure for the three main psychological constructs that were measured based on a broad variety of different indicators, i.e., vulnerability, resilience and social cohesion. In the second step, we aimed at identifying how these latent factors systematically changed over time during the COVID-19 pandemic in Berlin, focusing on three measurement time points: before the pandemic in January 2020 (T1), during the first lockdown in March/April 2020 (T2) and after re-opening in June 2020 (T3). In the third and final step, we aimed at investigating individual differences and identifying the correlative interrelationships between these constructs and their changes over time.

## 2. Methods

### 2.1. Sample

This study reports on data that were assessed in the context of the CovSocial project, a longitudinal study that employs a multi-measurement approach to investigate biopsychosocial dimensions of vulnerability, resilience and social cohesion as impacted by the COVID-19 pandemic in 2020 and 2021 in a Berlin population (see Appendix A for a detailed project description). The data used in this study refer to the three timepoints in January 2020 (before the COVID-19 pandemic), March/April 2020 (during the first lockdown of the COVID-19 pandemic) and June 2020 (after restrictions were loosened in Berlin). Further longitudinal data over the course of one year as well as trait measures and genetic markers will be presented in separate papers of the CovSocial project.

The study sample includes 3522 participants between 18 and 65 years old (mean age = 43.95 ± 12.69 years, 65.11% female). It was characterized by an average of 17.04 ± 3.86 years of education, an average monthly net household income of EUR 3227 ± 1210 and a 55% full-time employment rate at the time of data assessment. Lifetime prevalence of diagnosed mental disorders was reported by 24.87% (*n* = 95 non-disclosed responses). Of the total participants, 36.97% were married and cohabiting. Moreover, 23.74% of participants were considered physically at risk of COVID-19 due to medical conditions, and 24.56% of participants were working in a profession with increased risk of COVID-19. Please see Appendix A for detailed information about all sample demographics and their representativeness of the Berlin population, as well as descriptive statistics on trait measures assessed in the context of the CovSocial project.

Participants were recruited using various recruitment methods, including 56,000 letters to addresses that were randomly selected by the residents’ registration office in Berlin, e-mail lists of academic and research institutions, flyers at churches and sports clubs, social media postings, as well as advertisements in newspapers and on public transportation. Out of 7214 participants who signed up for the CovSocial project, 3681 participants completed the first seven blocks of self-report surveys on the project’s online survey platform (see Appendix A for a detailed report on dropout rates per survey block). We further excluded participants who did not meet the inclusion criteria, *n* = 44 non-Berlin residents and *n* = 81 individuals who were not between 18 and 65 years of age. Additionally, speed thresholds were defined for each survey block to flag participants with response times that were considered too fast to be reliable. To this end, five staff members (mean age 23.8 ± 2.77 years) who were highly familiar with the questions responded to all questions in a speedy manner while maintaining meaningful response patterns. The fastest response time was selected as the speed threshold for each survey block, and participants with speed flags in at least two of the seven blocks (*n* = 30) were excluded from the final sample.

The study is in accordance with the Declaration of Helsinki and received approval by the ethical committee of the Charité—Universitätsmedizin Berlin (#EA4/172/20). All study participants provided written informed consent. While no direct financial compensation was offered, five tablets were raffled using random selection among those participants who completed all seven survey blocks.

### 2.2. Study Design

Data were assessed using the project’s online survey platform (www.covsocial.de). The surveys were presented in seven blocks, with the first block consisting of demographic variables and the second, fourth and sixth block consisting of retrospective assessments of subjective perceptions during the pandemic with reference to three different time periods. Accordingly, the second survey block referred to January 2020 (T1); the fourth block to mid-March to mid-April 2020 (T2), the time of the first country wide COVID-19 pandemic-related lockdown; and the sixth block referred to June 2020, after the first lockdown (T3). The remaining third, fifth and seventh survey blocks included questionnaires on individual trait characteristics. Each participant was given four weeks to complete all survey blocks with the possibility to log in and out of participant accounts implemented on the web application of the CovSocial project. The assessment period spanned from 11 September 2020 to 7 December 2020.

### 2.3. Measures

State measures of vulnerability, resilience and social cohesion mainly consisted of self-generated questions, developed to assess time-varying pandemic-related aspects of these constructs. Self-generated questions were evaluated on 9-point Likert scales unless stated otherwise. The entire list of self-generated questions can be found in the Appendix A. Additionally, these state measures were complemented by several validated scales.

#### 2.3.1. Vulnerability

State vulnerability was assessed through 18 indicators on multiple areas of general as well as pandemic-specific vulnerability. General vulnerability included questions about aggression as perpetrator, aggression as victim, alcohol consumption, alcohol loss of control, craving for internet activities, internet use, perceived stress, overall stress, depressive symptoms, anxiety symptoms, overall anxiety, loneliness and psychosomatic complaints. Pandemic-specific vulnerability referred to pandemic-related fears, pandemic-related behaviors, protection measures and news consumption.

The experience of verbal and physical aggressive behavior was assessed using seven self-generated items each with regard to aggression as perpetrator or aggression as victim. Alcohol consumption was assessed using the consumption items of the Alcohol Use Disorder Identification Test (AUDIT-C; [104]). The AUDIT-C has been widely used as a screening instrument for problematic alcohol use. It consists of three items regarding the frequency and magnitude of alcohol consumption on a 5-point rating scale. Item scores were summed to an overall score of alcohol consumption. Additionally, one self-generated item was included to measure perceived loss of control over alcohol consumption. Compulsive internet consumption was assessed using the short form of the Compulsive Internet Use Scale (CIUS-5; [105]). The CIUS-5 entails five items that reflect clinical criteria for pathological internet use and gambling on a 5-point rating scale. A sum score across all five items was generated. Seven further self-generated items assessed cravings for internet activities, including online gambling, shopping, videogaming, streaming of movies or series, use of social networks, pornography or other types of internet surfing. To measure the degree to which life circumstances are perceived as stressful, the short version of the perceived stress scale was used (PSS-4; [106]). Four items on a 5-point rating scale assess how predictable, controllable and overwhelming situations were perceived during the time period in question. A perceived stress score was obtained by reversing the scores of the two positively phrased items and summing across all four items of the PSS-4 [106]. In additional, a self-generated item assessed the degree of perceived overall stress. Depressive symptoms were assessed using the two-item Patient Health Questionnaire (PHQ-2; [107]). The two items refer to curbed interest in activities and depressed mood on a 4-point rating scale. The PHQ-2 score is derived by summing up both items. Similar to depressive symptoms, symptoms of anxiety were measured with the brief two-item version of the generalized anxiety disorder scale (GAD-2; [108]). The items represent core symptoms of anxiety disorders on a 4-point rating scale, and are summed to form an overall GAD-2 score. We further included one item each to measure the perceived level of overall anxiety and loneliness. Seven self-generated items assessed psychosomatic symptoms, including back pain, fatigue, sleep disturbances, headache, cardiovascular or gastrointestinal diseases and symptoms of a cold.

Pandemic-related fears such as running out of food, toilet paper, disinfectants or money, and also fears of job loss, infection with diseases or viruses, an overloaded health care system, political or economic crises, international conflicts and an overall threatened existence were assessed on 11 items. Pandemic-related behaviors such as stocking up food, toilet paper, disinfectants or money were measured on four items regarding their frequencies. The subjectively perceived appropriateness of pandemic-related protection measures was assessed through four items, and the frequency of negative news consumption through one item.

Additionally, the questionnaires included 24 items on perceived strains and burdens that might be relevant in general and in the context of the COVID-19 pandemic, e.g., economic burdens, childcare or living conditions, among others. Items were rated regarding the extent that participants felt affected by them.

#### 2.3.2. Resilience

Self-generated resilience measures included one item each on speed and ease of stress recovery. Moreover, participants were asked whether they perceived crises as a chance for either themselves or for society in two items. Additionally, one item each was included to assess general optimism and life satisfaction. The assessment of coping with difficult situations entailed 12 self-generated items, each of which represented a different coping approach, namely, stress relief by social support, exercise, humor, religion, culture, music, nature, acceptance of the situation, active efforts to change the situation, distraction, release of emotions and planning. Self-efficacy beliefs were measured using the General Self-Efficacy Short Scale (ASKU; [109]). The scale consists of three items on expected competencies in planning and executing behavior to successfully achieve a desired goal. The appraisals were rated on a 5-point scale and the mean across the three items was calculated.

#### 2.3.3. Social Cohesion

All items that were used to measure dimensions of social belonging, trust, social interactions and prosocial behavior were developed to reflect the multi-system nature of the construct of social cohesion.

Feelings of social belonging were rated using the Inclusion of Other in the Self scale (IOS scale; [110]). The IOS scale consists of seven pairs of circles. In each pair, one circle is labeled “self” and the other one is labeled with a group of people (i.e., “family”, “friends”, “neighbors”, “Berlin”, “Germany”, “Europe”, “world”). Participants are requested to indicate the amount of social belonging to the respective group by using a slider to change the distance between the circles. The distance ranges from far away (=0) to completely overlapping (=100).

The extent of trust in a group of others was rated separately for family, friends, neighbors, the community, public media, the police, the Senate of Berlin, the German chancellor, the government, the health system and the economy.

Social interactions were assessed regarding their frequency and quality. Thereby, social contact in person and online contact were assessed separately. Due to the impact of the COVID-19 pandemic, particularly of the pandemic-related lockdown, on the frequency of social interactions, online contact was excluded from data analyses in the current study, while frequency and quality of social interactions in person were accounted for. Seven items each assessed frequency and quality of interactions with seven different types of people or groups of people, including partners, family members, friends, neighbors, colleagues, supervisors or others.

Prosocial behavior and experience were assessed in relation to the same seven person groups that participants were asked to rate regarding frequency and quality of social interactions. The frequency of prosocial acts from oneself to others and from others to oneself was assessed in two separate items for each type of person or group. Additionally, a self-generated item measured prosocial behavior with regard to temporal efforts. The frequency of social participation and political participation were each measured with one item.

### 2.4. Data Analysis

Statistical data analyses were conducted in three steps, including factor analyses, measurement invariance analyses and latent change score (LCS) analyses. All analyses were performed in R (version 3.6.3; [111]) (using a structural equation modeling framework as implemented in the lavaan package [112]. A significance level of *α* = 0.05 was used for all analyses.

Preprocessing steps included tests for the normality of sample distributions, using Shapiro–Wilk tests for univariate normality and Mardia’s test for multivariate normality. Self-generated items were rescaled to values from 0 to 8, since the majority of items were scaled accordingly. Internal consistencies for validated scales as well as for pools of self-generated items that were assumed to relate to the common overarching psychological dimensions outlined above were determined using Cronbach’s alpha. For self-generated questions, composite scores were computed by averaging across items when Cronbach’s alpha exceeded a threshold of 0.70 (Appendix A). For multi-system indicators of social cohesion on micro (e.g., person, families, partners, friends), meso (e.g., institutions, city) and macro levels (e.g., nation, international, global), no composites were computed regardless of item consistencies across the respective dimensions.

#### 2.4.1. Factor Analyses for Single Constructs

In the first step of analyses, confirmatory factor analyses (CFA) for each of the three constructs of psychological vulnerability, resilience and social cohesion were conducted separately. For the purpose of model identification, variances of the latent factors were constrained to 1, and means of latent factors were constrained to 0, while factor loadings, error variances and intercepts of manifest variables were estimated.

Due to the multi-system nature of social cohesion indicators, specific items on social interactions and social engagement were not applicable to some of the participants, and were rated as such. Particularly, items on social interactions with colleagues or supervisors and with partners showed the highest rates of not being applicable consistently across the three time periods T1, T2 and T3, while interactions with friends showed the lowest rates (see Appendix A). Items with missing data in >20% of cases were excluded from further analyses due to possible effects on parameter estimates, standard errors and model comparison [113]. In all SEM models with missing data, we used full information maximum likelihood (FIML) for parameter estimation, since this method showed superior performance over other estimation techniques in SEMs under the assumption that missing values are missing at random (MAR) [114]. To account for non-normality in item and scale distributions, robust maximum likelihood estimation [115] and robust indices of model fit, including root mean square error of approximation (RMSEA), the comparative fit index (CFI) and the Tucker–Lewis index (TLI), were used. Model fit was considered acceptable with RMSEA < 0.10 and CFI and TLI > 0.90. RMSEA values are reported with 90% confidence intervals. Moreover, for each factor model, chi-square (*χ*^2^) statistics and degrees of freedom (*df*) are reported. In case of insufficient model fit of the three constructs, exploratory factor analyses (EFA) as well as model modifications based on modification indices obtained using Lagrange Multiplier Tests were conducted. Preconditions for the computation of EFAs were tested using Bartlett’s test of sphericity for all measurement occasions, Kaiser–Meyer–Olkin criterion (KMO) > 0.50 and Measure of Sample Adequacy (MSA) <0.50. EFAs were based on parallel analyses and eigenvalues >1. A two-stage promax rotation was implemented to determine factor loadings of observed variables. Promax rotation maximizes high- and low-value loadings in comparison to mid-value loadings and has evidenced accuracy in estimating factor structure with both correlated and uncorrelated factors [116]. In an exploratory approach for measuring social cohesion, residual variances of the indicators that relate to the same kind of social other (e.g., family, friends, etc.) were modeled as additional latent factors. For all exploratory analyses, hold-out cross validation was implemented using a randomly selected hold-out test sample of 50% (*n* = 1761) and fit indices of RMSEA < 0.10 and CFI and TLI > 0.90.

#### 2.4.2. Integrative Measurement Models

In the next step of data analysis, the three constructs of vulnerability, resilience and social cohesion were combined into an integrative measurement model. Model choice was dependent on the comparative fit of structural models as well as on parameter constraints of measurement invariance. With regard to factor structure, three different measurement models were compared with each other. Firstly, the pre-registered measurement model with three distinct yet correlated latent factors for vulnerability, resilience and social cohesion was computed. The second model was based on findings from the single-construct level. Thirdly, an alternative factor structure with a general factor for resilience and vulnerability indicators and an additional specific factor for resilience indicators was modeled on theoretical plausibility (Figure 1). For model identification purposes, factor means at T1 were constrained to 0 and factor variances at T1 were constrained to 1. The three models were compared using chi-square difference tests.

In order to be able to compare means of latent factors across time, required constraints of scalar measurement invariance were included in all integrative measurement models. Scalar measurement invariance is given when factor structure, factor loadings and intercepts of indicators are equal across all measurement occasions. In the case of second-order factor models, intercepts of first-order latent factors were set equal across measurement time points [117]. Error variances of items, composites and latent factors were allowed to covary across T1, T2 and T3 [118]. In case of insufficient model fit, partial invariance was tested by releasing some model constraints based on modification indices.

#### 2.4.3. Latent Change Score Modeling

In the LCS model, established measurement invariance constraints were kept as a prerequisite for meaningful change analyses. In addition, latent change score factors were defined for each construct. The amount of change in latent factors from T1 to T2 and from T2 to T3 is thereby indicated by the means of two distinct latent change score factors (LCS1 and LCS2) and inter-individual differences in change by the residual variance of respective latent change score factors [119]. LCS1 factors were further regressed on baseline scores of latent factors at T1, and LCS2 factors were regressed on latent factor scores at T2. Thus, the reported relations are controlled for the previous time point. This implies that the reported covariances are covariances of residuals. For the sake of simplicity, we hereafter refer to them as “covariances” without explicitly stating that they are covariances of residuals (same for derived statistics, e.g., correlations). Covariances between LCS1 and LCS2 were estimated and converted into correlations to assess the association of change between the different constructs and between subsequent patterns of change.

To investigate between-construct relationships, covariances and correlations were computed between all T1, LCS1 and LCS2 factors. To account for within-construct regressions in the visualization of between-construct correlations, residuals of extracted latent factor scores were calculated by multiplying regression coefficients (e.g., from the regression of adaptive coping LCS1 on adaptive coping latent factor at T1) with the corresponding predictor (e.g., adaptive coping latent factor at T1) and subtracting this term from the latent change score of interest (e.g., adaptive coping LCS1). Furthermore, correlation plots included a rescaling of LCS factors by mean latent change, so that scores above 0 indicate actual increase and scores below 0 indicate decrease. Furthermore, in an extended LCS model, time-lagged regressions between T1 and LCS1 factors and between T2 and LCS2 factors were included between constructs, which represent time-dependent effects of one construct on the subsequent change in the other constructs [120]. Standardized estimates of intercepts, variances, correlations and regression weights are reported with 90% confidence intervals.

## 3. Results

### 3.1. Factor Structure

To determine the structure of the latent constructs of psychological vulnerability, resilience and social cohesion, confirmatory factor analyses were conducted for each construct and each of the three measurement occasions (T1, T2 and T3) separately. Item statistics, including internal consistencies of validated scales and of pools of self-generated vulnerability and resilience indicators at each measurement timepoint, are reported in Appendix A. Based on internal consistencies of *α* ≥ 0.70, composite scores were built for all pools of self-generated items, except protection measures (Appendix A).

#### 3.1.1. Vulnerability

The CFA for a one-factor solution of vulnerability did not reach an acceptable fit for any of the three measurement timepoints (T1: CFI = 0.67, TLI = 0.64, RMSEA = 0.101 [0.099, 0.103]; T2: CFI = 0.67, TLI = 0.63, RMSEA = 0.114 [0.112, 0.117]; T3: CFI = 0.68, TLI = 0.64, RMSEA = 0.112 [0.109, 0.114]). Therefore, EFAs were computed for each measurement occasion separately (Bartlett’s test of sphericity for all measurement occasions < 0.001; T1: KMO = 0.85; T2: KMO = 0.87; T3: KMO = 0.85; MSA < 0.50 for T2: protection measures item 3: 0.463, protection measures item 4: 0.429, AUDIT-C: 0.493; MSA < 0.50 for T3: protection measures item 4: 0.458, AUDIT-C: 0.497). Based on parallel analysis and eigenvalue criterion >1, a five-factor solution was suggested for all measurement timepoints in the training sample (see Appendix A). This factor structure could be validated with CFAs in the 50% hold-out test sample (T1: CFI = 0.94, TLI = 0.92, RMSEA = 0.058 [0.053, 0.062]; T2: CFI = 0.92, TLI = 0.89, RMSEA = 0.072 [0.068, 0.076]; T3: CFI = 0.93, TLI = 0.91, RMSEA = 0.068 [0.064, 0.073]). The first factor that explained the largest proportion of variance across T1, T2 and T3 was formed by mental health status (GAD-2, PHQ-2), perceived stress (PSS-4, self-generated item), overall anxiety, loneliness, perceived burdens and psychosomatic complaints (see Appendix A). The eigenvalues of all other four factors were considerably lower and close to 1. Due to the representation of mental health status on this factor, it was considered to reflect core vulnerability. Only the core vulnerability factor will be included in further integrative analyses on vulnerability, resilience and social cohesion. When modeled separately, the core vulnerability factor had acceptable overall fit at all measurement timepoints (T1: *χ*^2^ = 270.79, *df* = 20, CFI = 0.95, TLI = 0.93, RMSEA = 0.095 [0.085, 0.105]; T2: *χ*^2^ = 473.92, *df* = 20, CFI = 0.94, TLI = 0.91, RMSEA = 0.120 [0.111, 0.129]; T3: *χ*^2^ = 355.94, *df* = 20, CFI = 0.95, TLI = 0.93, RMSEA = 0.106 [0.097, 0.116]).

#### 3.1.2. Resilience

The resilience model with a one-factor solution was not supported by the CFAs for any of the measurement timepoints (T1: CFI = 0.85, TLI = 0.75, RMSEA = 0.148 [0.139, 0.159]; T2: CFI = 0.90, TLI = 0.83, RMSEA = 0.137 [0.127, 0.146]; T3: CFI = 0.90, TLI = 0.84, RMSEA = 0.143 [0.133, 0.153]). Again, EFAs were computed for each measurement occasion separately (Bartlett’s test of sphericity for all measurement occasions < 0.001; T1: KMO = 0.74; T2: KMO = 0.78; T3: KMO = 0.80; MSA > 0.50 for T1, T2 and T3). Parallel analysis and eigenvalue criterion yielded a two-factor solution with life satisfaction, optimism, self-efficacy and stress recovery loading on one factor, and coping as well as perceiving crises as a chance on a second factor; the eigenvalue of the latter factor was close to 0 (see Appendix A). The first factor was considered to reflect core resilience. The two-factor solution had an acceptable model fit in the hold-out test sample (T1: CFI = 0.98, TLI = 0.96, RMSEA = 0.057 [0.042, 0.073]; T2: CFI = 0.96, TLI = 0.93, RMSEA = 0.083 [0.069, 0.098]; T3: CFI = 0.95, TLI = 0.91, RMSEA = 0.105 [0.090, 0.120]).

#### 3.1.3. Social Cohesion

The CFA with four first-order factors of belonging, trust, social interactions and social engagement and one second-order factor of social cohesion did not reach an acceptable model fit for any of the three measurement timepoints (T1: *χ*^2^ = 22,314.53, *df* = 556, CFI = 0.55, TLI = 0.52, RMSEA = 0.113 [0.112, 0.114]; T2: *χ*^2^ = 21,219.65, *df* = 556, CFI = 0.62, TLI = 0.60, RMSEA = 0.109 [0.108, 0.110]; T3: *χ*^2^ = 26,236.06, *df* = 556, CFI = 0.58, TLI = 0.55, RMSEA = 0.123 [0.121, 0.124]). Likewise, the model without the second-order factor did not reach an acceptable fit (T1: *χ*^2^ = 23,298.48, *df* = 560, CFI = 0.55, TLI = 0.52, RMSEA = 0.115 [0.114, 0.116]; T2: *χ*^2^ = 21,747.64, *df* = 560, CFI = 0.61, TLI = 0.59, RMSEA = 0.110 [0.109, 0.111]; T3: *χ*^2^ = 27,192.70, *df* = 560, CFI = 0.56, TLI = 0.53, RMSEA = 0.124 [0.123, 0.126]). Therefore, EFAs were computed separately for T1, T2 and T3 (Bartlett’s test of sphericity for all measurement occasions < 0.001; T1: KMO = 0.83; T2: KMO = 0.82; T3: KMO = 0.84; MSA > 0.50 for T1, T2 and T3). Parallel analysis and eigenvalue criterion suggested 7 factors for T1, T2 and T3. Results of EFAs highlight that particularly those items that measure belonging, trust, social interactions or social engagement with respect to the same social groups load on the same latent factor (see Appendix A). Items on the total amount of prosocial efforts, political participation and social participation consistently showed low factor loadings (<0.30) and were excluded from further analyses. However, CFAs based on these factors alone did not yield an acceptable model fit (T1: *χ*^2^ = 7692.79, *df* = 464, CFI = 0.70, TLI = 0.68, RMSEA = 0.101 [0.099, 0.103]; T2: *χ*^2^ = 8487.21, *df* = 464, CFI = 0.70, TLI = 0.68, RMSEA = 0.105 [0.103, 0.107]; T3: *χ*^2^ = 9030.67, *df* = 464, CFI = 0.71, TLI = 0.69, RMSEA = 0.109 [0.107, 0.111]). Therefore, in accordance with the previously suggested multidimensional and multilevel nature of the concept of social cohesion [21], an extended hierarchical measurement model was proposed, not only including the four first-order factors belonging, trust, social interactions and social engagement and one second-order general factor of social cohesion, but also five additional residual factors that subsume items with regard to their reference to different social groups, i.e., family, friends, neighbors, others and institutions (Figure 2; see Appendix A for factor loadings). CFAs showed acceptable model fit for all three measurement timepoints (T1: *χ*^2^ = 5286.37, *df* = 428, CFI = 0.90; TLI = 0.90; RMSEA = 0.060 [0.059; 0.062]; T2: *χ*^2^ = 5473.22, *df* = 428, CFI = 0.91; TLI = 0.90; RMSEA = 0.060 [0.059; 0.062]; T3: *χ*^2^ = 5723.72, *df* = 428, CFI = 0.91; TLI = 0.90; RMSEA = 0.063 [0.061; 0.064]).

#### 3.1.4. Integrative and Invariant Factor Structure

In an integrative approach, we first tested our pre-registered measurement model (Model 1), in which indicators of vulnerability, resilience and social cohesion were modeled as three distinct yet interrelated latent factors. The CFA to test this model with measurement invariance across T1, T2 and T3 included core vulnerability, core resilience and social cohesion factors that were derived on a single-construct level. This three-factor structure was not supported by model fit (*χ*^2^ = 60,363.14, *df* = 9204, CFI = 0.88; TLI = 0.87; RMSEA = 0.040 [0.039; 0.040]). Based on EFAs on a single-construct level reported above and theoretical plausibility, two further measurement models were tested. Model 2 entailed the four distinct latent factors found on a single-construct level—that is, core vulnerability, two resilience factors and one second-order social cohesion factor—as well as covariances between these factors. The model did not reach an acceptable model fit (*χ*^2^ = 59,711.64, *df* = 9188, CFI = 0.88; TLI = 0.88; RMSEA = 0.040 [0.039; 0.040]). It further revealed a strong negative correlation between core vulnerability and core resilience of *r* = −0.92 at T1, *r* = −0.96 at T2 and *r* = −0.94 at T3. Together with theoretical notions of a complementarity between vulnerability and resilience [51,52], they were modeled on one bipolar resilience-vulnerability factor in Model 3. In alignment with previous resilience frameworks, resilience indicators were additionally modeled on a separate factor (Figure 1), since it has previously been suggested that resilience entails distinct processes beyond the more general overlap with vulnerability, which can be referred to as adaptive coping [121]. The results of this model showed high factor loadings of stress recovery, life satisfaction, optimism and self-efficacy on the general resilience-vulnerability factor, and high factor loadings of coping and perceiving crises as a chance on the residual factor. Due to a significantly better model fit, *χ*^2^*_diff_* = 2571.2, *df_diff_* = 19, *p* < 0.001, and theoretical plausibility, we chose Model 3 for further analyses. Scalar measurement invariance was not acceptable with all variable constraints, *χ*^2^ = 55,467.49, *df* = 9197, CFI = 0.89; TLI = 0.89; RMSEA = 0.038 [0.037; 0.038]; however, partial scalar measurement invariance was given when constraints on first-order factor intercepts of social engagement, and intercepts of method factors family, neighbors and others were released at T1, as suggested by modification indices derived from Lagrange Multiplier Tests, *χ*^2^ = 52,561.94, *df* = 9193, CFI = 0.91; TLI = 0.90; RMSEA = 0.037 [0.036; 0.037].

### 3.2. Within-Construct Changes and Predictions of Change

The LCS model (Figure 3) had an acceptable model fit (*χ*^2^ = 47,177.21, df = 9228, CFI = 0.91, TLI = 0.91, RMSEA = 0.034 [0.034, 0.034]). There was a significant mean increase in psychological vulnerability from T1 to T2 (LCS1 mean = 0.72 [0.68; 0.76]) and a significant mean decrease from T2 to T3 (LCS2 mean = −0.66 [−0.71; −0.61]). With regard to adaptive coping, there was a significant mean decrease from T1 to T2 (LCS1 mean = −0.33 [−0.43; −0.23]) and no significant change from T2 to T3 (LCS2 mean = 0.08 [−0.01; 0.17], *p* = 0.068). Social cohesion was found to significantly decrease on average from T1 to T2 (LCS1 mean = −0.86 [−0.99; −0.72]) and to increase on average from T2 to T3 (LCS2 mean = 0.73 [0.57; 0.89]). T3 levels on all three factors did not return to T1 baseline levels (Figure 4). Variances of all latent change scores were significant.

Results also show significantly negative predictions of latent factors at T1 on respective change scores from T1 to T2; that is, *β* = −0.25 [−0.29; −0.21] for resilience-vulnerability, *β* = −0.31 [−0.38; −0.25] for adaptive coping and *β* = −0.26 [−0.33; −0.19] for social cohesion. Moreover, for resilience-vulnerability, factor scores at T2 could negatively predict LCS2, *β* = −0.08 [−0.15; −0.01]. For social cohesion, factor scores at T2 could positively predict LCS2, *β* = 0.20 [0.10; 0.31]. Adaptive coping factor scores at T2 could not predict changes in adaptive coping from T2 to T3. Correlations between LCS1 and LCS2 were significantly negative for resilience-vulnerability, *r* = −0.42 [−0.49; −0.35], adaptive coping, *r* = −0.28 [−0.43; −0.13], and social cohesion alike, *r* = −0.36 [−0.49; −0.22].

### 3.3. Between-Construct Interrelations and Predictions of Change

Correlations between the three latent factors at T1 were significantly negative between resilience-vulnerability and social cohesion, *r* = −0.39 [−0.43; −0.35], and significantly positive between social cohesion and adaptive coping *r* = 0.54 [0.49; 0.59]. With regard to a relationship between the change scores of the three latent factors from T1 to T2, there was a significantly negative correlation between changes in resilience-vulnerability and changes in both adaptive coping, *r* = −0.31 [−0.40; −0.21], and social cohesion, *r* = −0.20 [−0.25; −0.15]. A higher increase in vulnerability was thereby correlated with a higher decrease in adaptive coping and social cohesion. The correlation of changes in adaptive coping and social cohesion from T1 to T2 was significantly positive, *r* = 0.37 [0.30; 0.44], in that a higher decrease in adaptive coping was associated with a higher decrease in social cohesion (Figure 5). Similarly, changes in resilience-vulnerability and both adaptive coping, *r* = −0.20 [−0.29; −0.10], and social cohesion, *r* = −0.14 [−0.20; −0.08], from T2 to T3 were marked by a significantly negative correlation, yet in reverse, a higher decrease in vulnerability was associated with a higher increase in adaptive coping and social cohesion (Figure 5). There was no significant correlation between changes in adaptive coping and social cohesion from T2 to T3.

The extended LCS model including time-lagged between-construct regressions had an acceptable model fit (*χ*^2^ = 43,686.40, df = 9200, CFI = 0.92, TLI = 0.92, RMSEA = 0.033 [0.032, 0.033]). Time-lagged regression paths show a significantly positive effect of social cohesion at T1 on subsequent latent change in resilience-vulnerability from T1 to T2, *β* = 0.13 [0.05; 0.21], and a significantly negative effect of social cohesion at T2 on resilience-vulnerability from T2 to T3, *β* = −0.16 [−0.25; −0.07]. None of the other time-lagged regressions reached statistical significance (Figure 6).

## 4. Discussion

The current study conducted in the context of the CovSocial project reports on a systematic investigation of the relationship between psychological vulnerability, resilience and social cohesion in the context of the COVID-19 pandemic in a large heterogeneous sample of Berlin inhabitants between 18 and 65 years of age. The investigation was based on a retrospective longitudinal design including a broad range of self-generated indicators and relevant validated scales assessed before (January 2020), during (March/April 2020) and after (June 2020) the first lockdown in Berlin, Germany.

In the first step, the best-fitting factor structures for the three main constructs of vulnerability, resilience and social cohesion were explored on a single-construct level, and subsequently in an integrative approach, probing a three-factor structure against alternative empirically driven and theoretically plausible models. Secondly, a latent change score model helped identify pandemic-related changes in these latent factors. Thirdly, based on the latent change score model, we explored the interrelationship between lockdown-related changes of psychological vulnerability, resilience and social cohesion, as well as their prediction. Thus, due to pervasive burdens of the ongoing pandemic on mental health, there is an increasing need for understanding resilience processes to sustain and recover mental health and strengthen psychosocial well-being. Furthermore, due to the specific requirements of social distancing throughout the pandemic, the study addressed whether social cohesion parameters similar to resilience could help buffer the detrimental effects of such a collective stressor, or whether the fabric of social life suffered a similar decline as mental health during these times of social isolation.

### 4.1. Factor Structure

In the first step, factor analyses on the single-construct level of 18 vulnerability indicators ranging from measures of stress, burdens, loneliness, anxiety or fears to maladaptive behavioral stress responses such as alcohol, internet or excessive news consumption and aggression, as well as mental health measures of depression and anxiety symptoms or psychosomatic complaints, were found to load on five different factors instead of one factor consistently across all three measurement occasions. Thereby, the factor that explained the most variance—that is, 21% (at T1) to 25% (at T3) of the total variance—included mental health scales on anxiety and depressive symptoms, perceived stress, overall anxiety and stress, perceived loneliness, psychosomatic complaints and perceived burdens. Due to the integration of core vulnerability constructs such as perceived stressor load [28], mental health [15] and loneliness [7,88] in this factor, it was considered to represent core psychological vulnerability. This is in accordance with the conceptual approach to vulnerability as an umbrella term for diverse stress responses, which are indicative of transdiagnostic dysfunctions in the mental health domain [43]. The other four smaller factors comprised indicators for aggressive behaviors, specific pandemic-related fears and behaviors, and alcohol or internet addiction. These factors are beyond the scope of this study and will be addressed in more detail in future analyses and papers focusing on pandemic-related changes in addictive and antisocial behaviors.

With regard to resilience, a two-factor solution emerged consistently across all three measurement occasions on a single-construct level, of which one factor was formed by the two indicators of coping during crises and perceiving crises as a chance. Comparative analyses of integrative measurement models, however, suggested a second factor that was formed by residual variance of all resilience indicators, which we elucidate in the following. In contrast to the proposed integrative factor structure, psychological vulnerability and resilience did not appear to be two distinct latent factors, yet respective indicators formed one general bipolar resilience-vulnerability factor with negative factor loadings of resilience indicators and positive factor loadings of vulnerability indicators (see Figure 1). However, variance of resilience indicators that was not explained by this general factor converged to an additional factor. Particularly high factor loadings on this specific factor were found for two indicators, namely, coping during crises and perceiving crises as a chance. In sum, and similar to a previous study performed in the CovSocial project focusing only on trait-level questionnaires [69], the best-fitting model incorporating multiple state indicators of vulnerability and resilience revealed one bipolar resilience-vulnerability factor and one residual resilience factor reflecting adaptive coping capacities not captured by core resilience characteristics loading on the vulnerability factor.

An interpretation of both the general and specific factor was based on the conceptual distinction of reactive versus (pro)active aspects of resilience [62]. Reactive resilience describes approaches that promote stability, while active resilience facilitates adaptation and transition, and is related to the human capacity for anticipation and learning [57]. This conjunction of resilience and adaptability has increasingly been proclaimed as a key aspect of resilience frameworks, since adaptability or adaptive, active coping leads to long-term and sustainable adjustments [43,55,56,123]. Adaptive, active coping can be characterized by changes at behavioral, neural, molecular and hormonal levels [62]. In line with these considerations, the general resilience-vulnerability factor was considered to reflect rather reactive resilience processes, while the specific factor was considered to reflect active resilience processes, including an investment of cognitive efforts to re-evaluate the stressor impact, and was therefore termed adaptive coping. This factor structure found on the level of state measures corroborated findings of the relationship between resilience and vulnerability based on trait measures in the same sample [69].

The construct-specific factor analyses focusing on social cohesion indicators revealed that as proposed, social cohesion could indeed be measured on the four psychological dimensions of social belonging, trust, social interactions and social engagement with respective indicators [20,74,75]. These four dimensions further loaded on a more general second-order social cohesion factor. Such findings are thus in line with early models of intelligence research, where different sub-components of intelligence such as speed, fluid and crystallized intelligence represent distinct abilities despite their marked interrelation, meanwhile converging on a more general intelligence factor, the so-called general g [83]. Similarly, a recent empirical framework of social cohesion that takes into account both its multidimensional and multilevel structure could confirm the existence of a second-order social cohesion factor, which is composed of several distinct dimensions [21]. Interestingly, our model did not reach acceptable model fit unless residual variances were grouped with reference to distinct social groups or entities (i.e., family, friends, neighbors, others, institutional, national or international entities) (see Figure 2). Therefore, on a psychological level, the concept of social cohesion can not only be measured on the dimensions of belonging, trust, social interactions and social engagement, but people also seem to have distinct response tendencies with regard to the group or entity they refer to. Beyond methodological aspects, this may relate to the notion of micro, meso or macro systems of social networks [25] and could indirectly be reflective of dimensions of social inclusion [21]. Further analyses will be needed to elucidate pandemic-related effects on inter- and intraindividual changes in these multi-system factors of social cohesion during the pandemic. For the present analyses focusing on the interrelationship between the core constructs of vulnerability, resilience and social cohesion over time, we chose to move forward with using the general social cohesion factor to reduce complexity. Future studies in the CovSocial project will aim at analyzing in more depth how this complex structure of social cohesion and its multiple sub-components are influenced by multiple demographic, context and trait factors over time.

In conclusion, here we could identify a reliable structure that reflects different aspects of social cohesion (trust, belonging, social interaction and social engagement) based on psychological subjective self-report measures, which could be validated to be stable over three different time points. As a result, the construct of social cohesion, which has mostly been discussed in social, political and economic science in the recent years, has been made accessible to the field of social and personality psychology and can be related to more established constructs such as resilience, vulnerability or other social and personality characteristics.

### 4.2. Within-Construct Changes and Predictions of Change

In the second step, after having established reliable factor structures of the three main constructs, we investigated retrospectively perceived changes in the latent factors of resilience-vulnerability, adaptive capacities and social cohesion during the COVID-19 pandemic, particularly from before the pandemic in January 2020 (T1) to the first lockdown in April/March (T2) and from lockdown to after re-opening in June 2020 (T3). Findings of latent change score analyses highlight that a collective lockdown effect took place in Berlin during the first pandemic-related lockdown in April/March 2020 (see Figure 4). Compared to baseline in January 2020, many aspects of psychological vulnerability such as perceived anxiety, loneliness, stress, depressiveness, perceived burdens and psychosomatic symptoms significantly increased in retrospect in April/March 2020 in the Berlin population. By nature of the model, indicators of resilience, which load on the same general factor as vulnerability indicators yet with opposite signs, and which represent protective capacities such as optimism, life satisfaction, stress recovery or self-efficacy beliefs, all significantly decreased during the first lockdown as compared to before the pandemic. Thus, these findings clearly speak to the fact that the lockdown was experienced as a collective stressor and shock by the majority of the Berlin population and lead to a perceived increase in mental health challenges and burdens and a concomitant decrease in adaptive capacities and resilience. This result extends previous findings on mental health impacts of the pandemic-related lockdown in Germany, which measured mental health status using one single scale [36] or several specific scales [37], to a much broader range of maladaptive and resilient stress responses on a more reliable latent level.

Interestingly, a similar time course was found for perceived adaptive coping capacities and social cohesion, which both significantly decreased from pre-pandemic baseline to the first lockdown (T1–T2). Thus, in contrast to the “tend-and-befriend” hypotheses and earlier accounts suggesting increased social cohesion after natural disasters [87] or other stressors [84,85], it seems that the specific nature of the pandemic-related lockdowns, which were associated with social distancing and often with social isolation and restrictions on engaging in social interactions and behaviors, actually led to a perceived decrease in social cohesion. This is consistent with other findings in the context of the first COVID-19 pandemic-related lockdown [13]. Taken together, our results emphasize that the lockdown negatively affected not only the mental health of Berliners but also resilience and social capacities that usually help buffer adverse effects of stressors.

After the lockdown, participants recovered from this shock, and vulnerability parameters decreased again at re-opening in June 2020, while social cohesion increased. However, this perceived recovery did not reach baseline levels from before the pandemic, and Berliners found themselves more vulnerable and less resilient and socially cohesive than at the beginning of the year (see Figure 4). Even though adaptive coping capacities slightly increased after the lockdown as well, this change was not significant.

Our findings are in contrast with other studies in German samples, which found no change in mental health status due to the lockdown [37] or even improved mental health after the lockdown [36]. This might be either due to differences in sample characteristics such as better mental health status [36] or higher levels of education in those studies [36,37]. About 25% of participants in the current sample reported having been diagnosed with a mental disorder in the past, which is comparable to the prevalence of mental disorders of approximately 28% of the general population in Germany [124], and these preconditions may relate to a stronger mental health impact of the pandemic [66]. Moreover, the CovSocial sample was rather heterogeneous and representative in terms of education, income (a bit higher than the Berlin population) and age (see Appendix A). The discrepancy with other German COVID-19 studies could also be due to differences in outcome measures, since our approach to psychological vulnerability as a construct of importance to mental health indeed goes beyond clinical symptomatology and also includes many sub-clinical indicators as well as stress-related and loneliness measures. It is noteworthy that despite significant observed average changes in resilience-vulnerability, adaptive coping and social cohesion factors between T1, T2 and T3, we also observed huge interindividual variability in these change patterns over the three measurement time points for all three constructs of interest. This is in line with previous findings in a German sample, which highlight that about 45% of variance in mental health during the first lockdown was explained by interindividual differences [36], and suggests that Berliners reacted to and recovered from this collective stressor in very different ways. Given this interindividual variability in shock response and recovery, as well as the heterogeneity of sample demographics, future studies in the context of the CovSocial project will address different trajectories of resilience and vulnerability over the course of the pandemic, and their prediction by demographic, individual trait characteristics and context factors.

### 4.3. Between-Construct Interrelations and Predictions of Change

In the third and final step, we focused on investigating how the three main constructs of social cohesion, adaptive coping and resilience-vulnerability were linked at the level of individual differences, particularly regarding their change over time (see Figure 5). In line with our prediction that social cohesion can help buffer against adversity and thus has resilient properties, we indeed observed that perceived adaptive coping and social cohesion shared significant variance before lockdown (*r* = 0.54). Furthermore, both change trajectories were aligned from baseline to the first lockdown, in that people who changed more in adaptive coping also changed more with respect to their social cohesion levels. Interestingly, however, this positive correlation was not observed anymore after re-opening in June.

Particularly, individuals whose psychological well-being and mental health were more strongly impacted by the lockdown—that is, who increased more in vulnerability—were characterized by stronger decreases in both adaptive coping capacities and perceived social cohesion as compared to before the pandemic (see Figure 5). These results suggest that resilience and social cohesion indeed have aspects in common, and that the specific nature of the lockdown as collective stressor has led to a general decrease in these potentially adaptive and buffering factors. It remains to be further explored which specific aspects of social cohesion as identified by the hierarchical measurement model are crucially driving this association with adaptive capacities, and are affected the most by the pandemic-related lockdown. Thus, while adaptive coping and social cohesion were coupled, they were not buffering vulnerability but rather declining together, suggesting that during the first lockdown, neither social cohesion nor adaptive coping could ward off an increase in vulnerability as predicted and instead collapsed. A reason for this may be that public health measures of social distancing and isolation prevented individuals from making use of social coping strategies when facing this unique type of stressor [91]. In line with this, previous findings have highlighted the malleability of social and psychosocial factors in the context of the COVID-19 pandemic, and their role in increasing the risk of mental health impacts [5,11].

Regarding interindividual variability after the lockdown, a higher decrease in vulnerability from lockdown to re-opening in June 2020 was indeed associated with a higher increase in adaptive coping capacities and social cohesion, suggesting that now individuals who could more adaptively employ coping skills and had higher levels of social interaction and engagement, trust and belonging to others after the stressor was over could recover more from its preliminary adverse effects. This is in line with previous research that highlights the resilience-promoting role of social connectedness and available social resources including social support [48,94,95,96,97]. Moreover, the strong associations between dynamic trajectories of psychological vulnerability and adaptive coping corroborate the notion that adaptive capacities, which are central to the concept of resilience, uniquely promote stress recovery [53,54]. In this sense, during stress recovery, adaptive coping regains its stress-buffering role as predicted.

The decoupling of adaptive coping and social cohesion after the lockdown may be traced back to the unique type of stressor that the pandemic-related lockdown represents, which is social by nature, since public health measures of social distancing and social isolation imposed massive restrictions on peoples’ social lives [91]. Changes in social cohesion might therefore be less indicative of the use of social resources as possible revenues related to adaptive coping, and therefore less internally but externally driven. Additionally, it is possible that specific first-order dimensions of social cohesion are more and others less associated with trajectories of adaptive coping capacities during crises, which is not reflected on the level of the more general second-order social cohesion factor. Drawing from a network perspective on community resilience, it was indeed found that communities characterized by strong bridging social capital are particularly adaptive [125]. Bridging social capital describes resources and processes that promote inclusion and tolerance in social relationships between groups [81,126], and has most conceptual overlap with the social cohesion dimension of social engagement and prosocial behavior. Whether or not trajectories of social engagement were more aligned with those of adaptive capacities in the context of the COVID-19 lockdowns remains an open question for future research endeavors.

Finally, besides the associations between synchronous change scores that are reported above, we investigated time-lagged change score correlations (see Figure 6); that is, how changes observed due to the first lockdown (T1-T2) relate to changes after re-opening (T2-T3). These results highlight a consistent pattern of oscillating stress responses of resilience-vulnerability, adaptive coping and social cohesion, which are in analogy with the concept of homeostasis, or self-regulating processes that promote a return to an approximation of a set point [127]. In this sense, the higher the deviance from a set point as impacted by the lockdown stressor, the higher the adjustment after.

In this sense, a higher increase in psychological vulnerability (and associated decrease in resilience) was associated with a subsequent higher decrease in those aspects after re-opening. Similarly, individuals who suffered more from reduced coping skills and social cohesion during the first lockdown increased more in these capacities after re-opening. This may speak to an initial shock response of individuals to an unpredicted collective stressor with a healthy recovery response afterward deploying the typically successful strategies of adaptive coping and seeking social support.

While previous research suggests that the experience of social cohesion can be a protective factor that promotes the maintenance of psychological well-being in the face of adversities in general [97,98,99], we observed that perceived social cohesion differentially predicted changes in resilience and vulnerability before as compared to during the lockdown. Although social cohesion and adaptive coping skills were indeed positively associated at baseline before the pandemic hit, higher pre-lockdown levels of social cohesion predicted a negative impact of the lockdown on markers of vulnerability and resilience, and thus, high social cohesion levels did not serve as a protective factor, but on the contrary increased the risk for psychological distress. Interestingly, other studies that found an association between social connectedness and lower levels of psychological distress in the context of the COVID-19 pandemic were either conducted in a cross-sectional design [96] or used pre-lockdown levels of perceived social support to predict mental health status after the lockdown [36]. In line with those studies, we also observed in our study that levels of social cohesion in April/March during the lockdown predicted better stress recovery after re-opening in June. Our findings thus highlight a unique and complex role of social cohesion at different stages of the pandemic, characterized by adverse repercussions that the lockdown had on individuals who were more embedded in their social environments before the pandemic. They also highlight the importance of maintaining or regaining perceived social cohesion despite enforced social isolation and distancing for the rebound of mental health and well-being. Together with the high interindividual variability of shock response and recovery, it remains an open question for future research whether different classes of resilient or vulnerable trajectories can be predicted by interindividual differences in social cohesion, and which specific dimensions of social cohesion might drive such developmental outcomes.

### 4.4. Limitations

One of the limitations of the current study is related to the retrospective assessment of the present longitudinal survey data. Given the unpredictable and unprecedented nature of the COVID-19 pandemic, a pandemic-related study could only be initiated once the pandemic had already hit. Therefore, retrospective study designs were a prevalent approach to investigate longitudinal mental health trajectories during the COVID-19 pandemic [29]. In the current study, data assessment could only take place several months after the first lockdown in Germany from 11 September 2020. However, to optimize the conditions for reliable and valid assessments of subjective perceptions before, during and after the first lockdown, we introduced text reminders to facilitate recall of the specific periods. More specifically, for each of the three time periods of interest (i.e., January 2020, mid-March to mid-April 2020 and June 2020), participants were presented with reminders of the political and societal situation in Berlin that was being reported in the news at the respective point in time in the form of short descriptive texts at the beginning of each survey block, and three additional periodical short prompts presented during the survey blocks. Time courses of pandemic-specific indicators such as pandemic-related behaviors and fears, which show very low values at T1, as well as responses to an additional question on how difficult it was to remember the specific time periods, which was asked in hindsight (mean = 2.92 ± 1.72, range = 0–8, higher scores represent more difficulties, missing *n* = 1049), revealed that participants could indeed distinguish between the specific time periods. This retrospective data assessment might have introduced some memory bias, particularly due to the ongoing pandemic at the time of assessment. However, the longitudinal assessments still offer insights into intra-individual change patterns and thus go beyond a purely cross-sectional understanding of the psychological impacts of the pandemic.

It is also noteworthy to mention that we used a rather broad definition of vulnerability and resilience. While in many other studies, the concept and thus measurement of resilience and vulnerability are defined by the presence or absence of mental health as an outcome [45], mental health status (e.g., depression) in this study is measured in the form of subclinical symptoms of anxiety and depression and is further merged with other aspects of psychological distress and dysfunction on the one hand and processes of self-regulation and coping on the other, resulting in one single resilience-vulnerability factor. Although this may be in contrast to several established frameworks, this approach was both theory- and data-driven. The COVID-19 pandemic represents a collective stressor that affected the entire population of our globe. Beyond mental health burdens, the lockdown-related restrictions, changes in daily routines such as home-office and home-schooling, limited access to medical services or other challenges inflicted stress and loneliness on people at a global scale. Based on our data, this broad range of vulnerability indicators including subclinical symptom levels of depression and anxiety indeed shared a lot of variance at baseline as well as during the pandemic. Furthermore, factor models with one single resilience-vulnerability factor had a better fit than the models postulating two unique factors for vulnerability and resilience. Future research using other longitudinal data sets in the context of the COVID-19 pandemic and outside of this particular context will have to establish whether such a conceptualization of vulnerability and resilience can be replicated.

## 5. Conclusions

Taken together, the current study focused on investigating the structural and temporal interplay between vulnerability, resilience, adaptive coping and social cohesion during the COVID-19 pandemic in Berlin using a retrospective longitudinal design and a broad range of markers of vulnerability and resilience as well as a novel approach to subjective psychological markers of social cohesion. This allowed us to identify a single factor with vulnerability and resilience forming two opposite poles of the same dimension. Interestingly, in addition to this core resilience-vulnerability factor, another unique factor emerged from resilience indicators reliably over the three time points, referring to adaptive coping strategies that allow one to buffer the detrimental effects of collective stressors such as a lockdown. Furthermore, our findings also advance the field of social cohesion by providing psychological questionnaire-based evidence for a hierarchical and complex construct composed of sub-components such as trust, belonging, social engagement and social interaction that are conceptually separable, as well as a second-level global social cohesion factor. This novel way of modeling social cohesion will provide a fruitful basis for future empirical investigations in the field of psychology, allowing us to link the more established constructs of resilience and vulnerability to the construct of social cohesion, which is mostly discussed in the social, political and economic sciences thus far.

Our results could further provide robust evidence for a general increase in vulnerability in the Berlin population during the first lockdown, with a slight recovery after re-opening. Presumably due to the specific nature of social distancing requirements during lockdown, high social cohesion levels, which were positively related to adaptive coping before the pandemic hit the Berlin population, also showed a decline during the first lockdown, and thus did not promote a stress-buffering effect. Similarly, adaptive coping capacities declined during the first lockdown, and even though they remained weakened, they could deploy their beneficial effects in buffering from adversity when Berlin eased their lockdown-related restrictions. Our results thus highlight the necessity to adopt a differentiated view of the specific conditions under which adaptive mechanisms such as the use of coping strategies, social skills and social support may have beneficial effects on stress recovery, while under certain circumstances such as under unexpected lockdowns they may even become risk factors for increased vulnerability. Furthermore, future analyses will have to focus on identifying which demographic, context and individual trait characteristics put certain individuals at higher risk of developing detrimental mental health trajectories which may be exacerbated by the cumulative effects of multiple lockdowns over several months. The COVID-19 pandemic indeed is a social crisis, not only on the level of the society, but also on the psychological level, and it is therefore of crucial importance to re-establish a sense of connectedness in societies and individuals alike.

## Figures and Tables

**Figure 1 ijerph-19-03290-f001:**
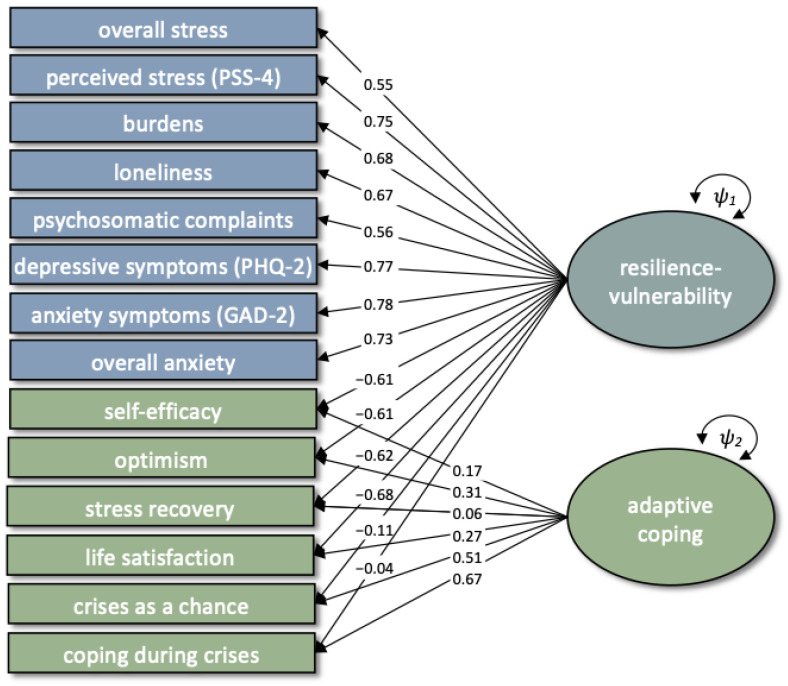
Measurement model with a bipolar resilience-vulnerability factor and a residual adaptive coping factor. Standardized factor loadings are reported. ψ = variance, box = observed variable, circle = latent factor. Mean structure-related model elements are not depicted. Measurement errors are included in the model but not displayed in the figure for reasons of clarity.

**Figure 2 ijerph-19-03290-f002:**
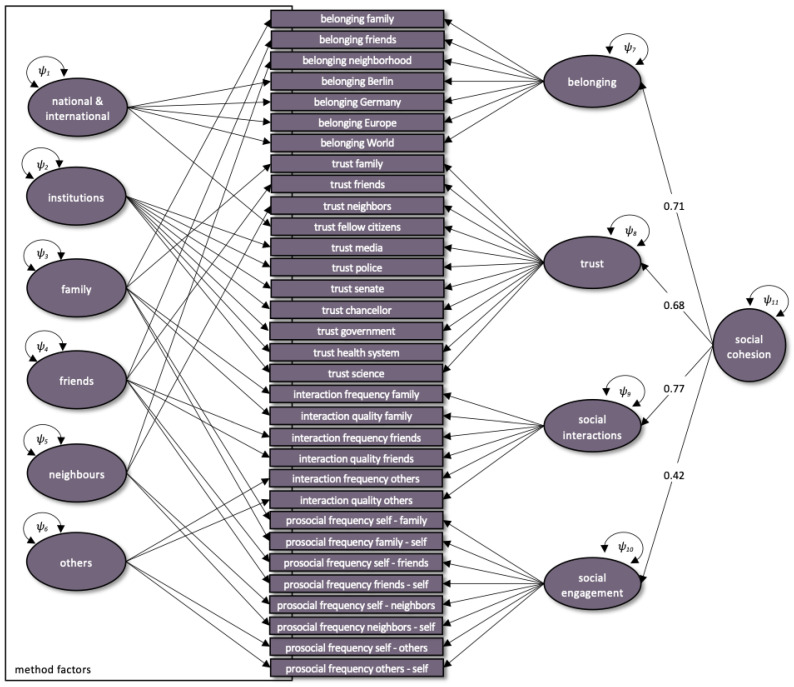
Hierarchical measurement model of social cohesion. ψ = variance, the variances of the first-order factors, ψ_7_–ψ_10_, are error variances; box = observed variable; circle = latent factor. Mean structure-related model elements are not depicted. Standardized second-order factor loadings are reported. Measurement errors are included in the model but not displayed in the figure for reasons of clarity.

**Figure 3 ijerph-19-03290-f003:**
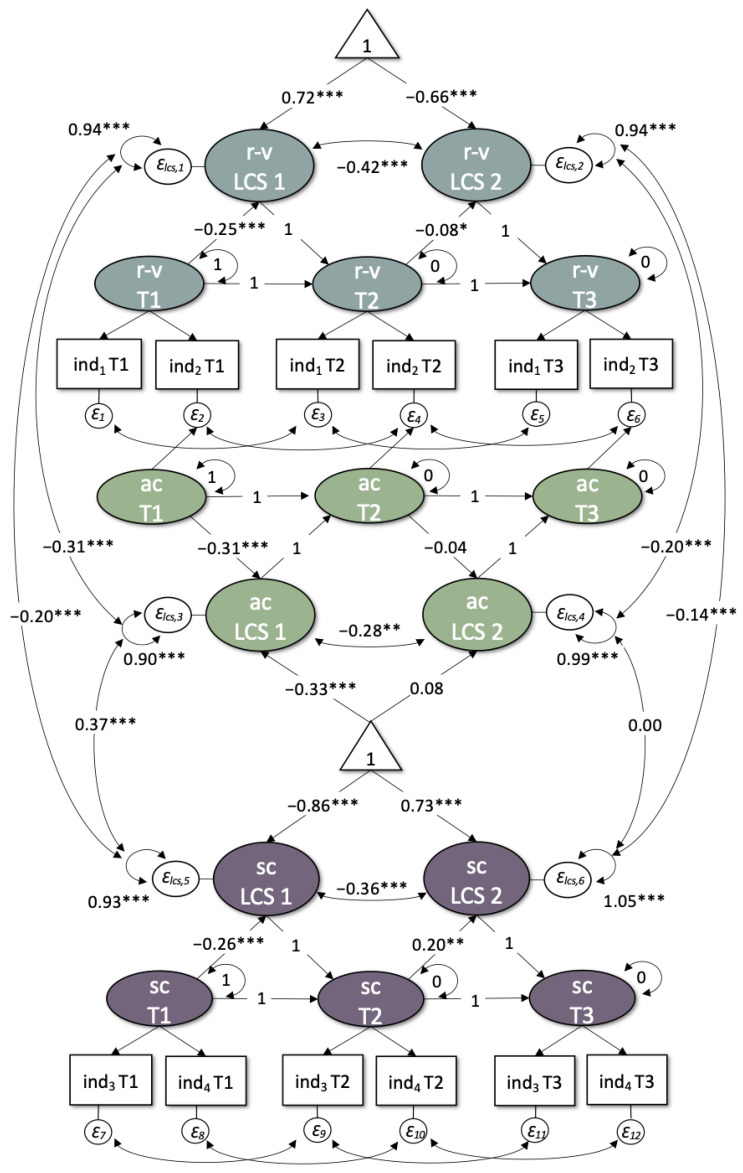
Latent change score (LCS) model with changes from baseline (T1) to the first lockdown (T2) and from lockdown to post-lockdown (T3) in resilience-vulnerability (r-v), adaptive coping (ac) and social cohesion (sc). Significance levels of * *α* = 0.05, ** *α* = 0.01 and *** *α* = 0.001.

**Figure 4 ijerph-19-03290-f004:**
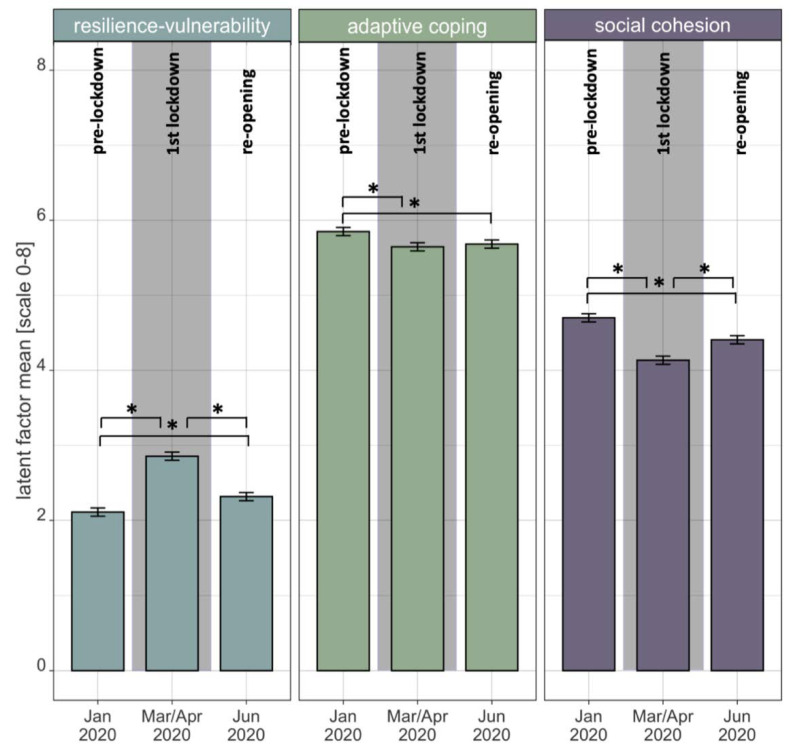
State change in resilience-vulnerability, adaptive coping and social cohesion as impacted by the COVID-19 pandemic-related lockdown. Means and 95% confidence intervals of latent factors are reported. For ease of interpretation, factors are scaled to marker indicators [122]. Significance level of * *α* = 0.001.

**Figure 5 ijerph-19-03290-f005:**
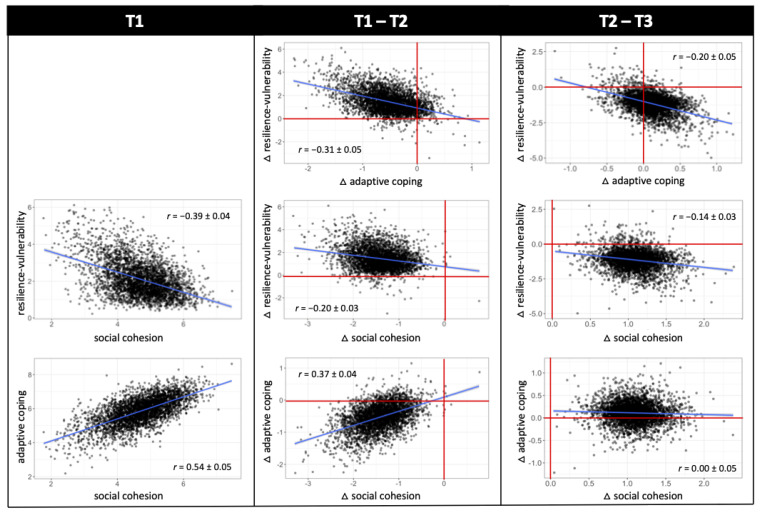
Scatterplots for the illustration of correlations between resilience-vulnerability, adaptive coping and social cohesion at baseline (T1) and of changes from baseline to the first lockdown (T2) and from lockdown to post-lockdown (T3). Negative change scores indicate a decrease, positive scores indicate an increase. The figure shows no correlation between resilience-vulnerability and adaptive coping at T1, because they are defined as orthogonal.

**Figure 6 ijerph-19-03290-f006:**
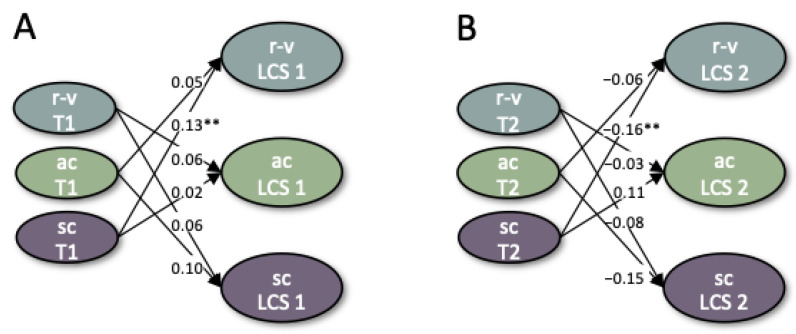
Between-construct time-lagged regression paths of (**A**) the relationship between baseline levels (T1) of resilience-vulnerability (r-v), adaptive coping (ac) or social cohesion (sc), and changes in those factors from pre-lockdown to lockdown (LCS 1); and (**B**) the relationship between lockdown state (T2) of resilience-vulnerability, adaptive coping and social cohesion, and changes in those factors from lockdown to post-lockdown (LCS 2). Significance level of ** *α* = 0.01.

## Data Availability

Data will be made available upon request.

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
