# Peer review of "Coping with the COVID-19 Pandemic: Perceived Changes in Psychological Vulnerability, Resilience and Social Cohesion before, during and after Lockdown"

_ijerph, 2022, doi:10.3390/ijerph19063290_

Round 1
Reviewer 1 Report
The present article presents a longitudinal study on the effect of Covid and the restrictions that have been set in Germany on individuals’ wellbeing, starting from the available literature on the pandemic. In particular, the focus is on the structural and temporal relationship between several psychological indicators such as resilience, vulnerability and social cohesion using a retrospective longitudinal design conducted on a sample of 3522 individuals.
The article is very well written and of sure interest, particularly in these days.
The introduction clearly presents the available literature on the topic and gradually introduces the reader to the objectives of the article.
The methods are also well described, and the statistical analyses (Factor Analyses and Latent Change Scores, which I liked as it fits well with the article objectives) correctly applied. Further, the results are clearly reported. By testing several models, the authors arrived at 3 general conclusions: i) vulnerability and resilience, rather than being two distinct factors, converged on a single, bipolar, latent variable; ii) another factor could be obtained originating from the residual variance of the indicators of resilience; iii) social cohesion was made of dimensions of trust, belonging, social interactions, and social engagement. Further, the authors also found that their LCS model showed that psychological vulnerability increased during the first lockdown and decreased at reopening, yet it did not go back to baseline levels. On the contrary, social cohesion decreased and then increased again. The effect of mental health was also more pronounced when pre-lockdown levels of social cohesion were higher.
The discussion section is also very clear, coherent, and structured. It starts by linking the present results to the available literature, then moves into interpreting the results in more details and discusses the limitations of the present study, such as the fact that their study was based on a retrospective design (i.e., asking information about past events, yet the authors reported that they dealt with this by sending participants reminders to facilitate the recalling of specific periods) and that they used a broad definition of vulnerability and resilience. So the authors did a good job in discussing the potential limitations of their study.
The supplementary material is also clear and complete.
In conclusion, I think that the quality of this study is high and that, in my opinion, the article could be accepted in the present form.
As a very minor point, I would suggest the authors to check their paper as there are a couple of points where the font/formatting changes (for example, line 72 and then again line 157). So, be careful that the document is correctly formatted throughout. Last, considering the high quality of the work, I would also suggest the authors to share their R code as an additional supplementary file so that other researchers could adopt a similar analytical strategy.
Author Response
We thank the reviewer very much for this encouraging feedback. We have corrected the errors in font and formatting in the revised manuscript. The R code will be shared with other researchers upon request.Reviewer 2 Report
Exciting research. I don't have any specific comments. Please check some typos and grammatical errors. Good luck
Author Response
We highly appreciate the reviewer's positive feedback on our manuscript.
We have proofread the manuscript and corrected some grammatical mistakes.
Reviewer 3 Report
The article entitled „Coping with the COVID-19 Pandemic: Changes in Psychological Vulnerability, Resilience and Social Cohesion before, during and after Lockdown” is an important source of knowledge for health psychology about the dynamics of selected health-promoting and anti-health psychological properties in a large group of 3522 examined during the crisis such as the COVID-19 pandemic. The manuscript is carefully prepared. You can see that the authors put a lot of work into its preparation. The presentation of both the review of the current literature and own research is prepared at a high level, with attention to accuracy. The article is certainly a very important source of knowledge that should be of interest to researchers around the world, because it provides new, interesting information and is inspiring.
The article is very extensive, especially in the introduction, which is usually a drawback of original works. In the case of the reviewed article, however, I believe that the introduction, although lengthy, is a very good study that should be read with interest.
A drawback of the article is the overload of data, especially numerical ones, in some of the results. There is a lot of numerical information in the text, which may make it difficult to focus on what is most important - the meaning of the results. However, I am aware that the indicated numerical values are important, and not everything can be included in more accessible figures.
Editorial errors have crept in several places - changing the font of the text, e.g. in lines 72-75 or 156-161, 165-167 – I suggest adjusting them.
Overall, I rate the quality of the presented manuscript highly. It is certainly a work that will be a valuable source of knowledge and inspiration for researchers interested in the phenomenon of psychological resilience. The extensive literature study also deserves attention.
Thank you for the opportunity to review this article and congratulations to the Authors on an interesting presentation of this research project.
Author Response
We thank the Reviewer very much for this encouraging feedback.
In our revised manuscript, we have divided the discussion into different subsections. These subsections are kept consistent with the those of the results section. We hope that this new structure will help the reader to more easily relate the vast amounts of reported data to meaningful interpretations and conclusions.
Reviewer 4 Report
In my view it is an article that is based on a very reflected and sustained work.
It has very strong strengths and some to improve. In more detail:
1. Title: As it is a study with a retrospective longitudinal design, I think it should be designated as the subjects' perception. Title suggestion: Perception of Psychological Vulnerability, Resilience and Social Cohesion before, during and after the Lockdown.
2. Summary: This is objective. It frames all the fundamental components. Keywords: lack COVID-19 pandemic.
3. Introduction: framed with current references and the relevance and objectives are explained.
4. Theoretical framework organized, well supported and framed. Well-presented state of the art, framing with current review and organized in a clear and objective way, definition of concepts (Psychological Vulnerability; Resilience and Social Cohesion) and interrelation between concepts.
5. Sample:
- Information on individual and sociodemographic characterization (e.g., years of schooling, profession, professional situation, family typology, with whom it lives, information on physical and psychological health status) is lacking, i.e., it would be important to describe demographic characteristics, individual traits and some more significant contextual factors.
- The age group is very wide (18-65 years). With the distinctions that characterize these periods of the life cycle so evident regarding the concepts analysed (vulnerability; resilience and social cohesion), I think it would be of central importance to carry out the analysis by age groups.
6. Study design- one of the limitations of this study is the fact that it has a retrospective longitudinal design - the perception may have been altered with the experience of the ansiogenic situation (world pandemic), since the data were collected between September 11th, 2020 to December 7th, 2020, should soon always approach as perception of the individual.
7. Measures- the concepts listed in the literature were evaluated and supported in validated instruments. The use of only some items of the scales, analysed through the means may have led to some biases. Important to explore the issue.
8. Data Analysis- A significance level of α = .05 was used for all analyses. Or p?
9. Conclusions- Repetitive and very extensive text that should be, in my point, divided into two or even three articles:
-Factor analyses - identifying and validating the most probable factor structure for the three main psychological constructs that were measured based on a broad variety of different indicators, i.e., vulnerability, resilience and social cohesion. Very interesting.
-Retrospective longitudinal design - identifying how these latent factors systematically change over time during the COVID-19 pandemic in Berlin (January 2020; March/April 2020 and June 2020. Although they use strategies to activate memories, we are talking about a very intense period in the psychoemotional organization, which brings high risks of bias. They should address/further explore this issue.
-Investigating individual and at identifying the correlative interrelationships between these constructs and their changes over time. These features are not explored in detail. It would be important to have more sample data and carry out their exploratory analyses.
Author Response
Dear Reviewer,
Thank you very much for your helpful comments and encouraging suggestions for our manuscript. We have made revisions accordingly. A point-by-point response is provided below.
Thank you very much for your time and consideration.
1. Title: As it is a study with a retrospective longitudinal design, I think it should be designated as the subjects' perception. Title suggestion: Perception of Psychological Vulnerability, Resilience and Social Cohesion before, during and after the Lockdown.
We changed the title to “Coping with the COVID-19 Pandemic: Perceived Changes in Psychological Vulnerability, Resilience and Social Cohesion before, during and after Lockdown”
2. Summary: This is objective. It frames all the fundamental components. Keywords: lack COVID-19 pandemic.
We added “COVID-19 pandemic” to the keywords.
3. Sample:
- Information on individual and sociodemographic characterization (e.g., years of schooling, profession, professional situation, family typology, with whom it lives, information on physical and psychological health status) is lacking, i.e., it would be important to describe demographic characteristics, individual traits and some more significant contextual factors.
In our first manuscript version, we referred to the supplementary material (Supplement 1) for further information on sample characteristics, including demographic variables and trait measures. We have now additionally included descriptive information on some demographic variables in the main manuscript.
- The age group is very wide (18-65 years). With the distinctions that characterize these periods of the life cycle so evident regarding the concepts analysed (vulnerability; resilience and social cohesion), I think it would be of central importance to carry out the analysis by age groups.
We thank the reviewer for this suggestion. We agree that it is an interesting research question to investigate differential impacts of the pandemic and pandemic-related lockdown in different age groups. We have assessed many demographic variables as well as COVID-specific context variables, which are all listed in Supplement 1 and of which several might be relevant predictors of differential impacts of the pandemic-related lockdown on resilience, vulnerability and social cohesion (for example, age, sex, socio-economic status or marital status, but also COVID-relevant health and living conditions).
Within the CovSocial project, there is an a priori publication agreement as a legally binding part of the cooperation contract between all cooperation partners and institutions (Humboldt-Universität zu Berlin, Max-Planck Society, Charité – Universitätsmedizin Berlin, Max-Planck, Charite) involved in this multimethod and multi-center CovSocial project. Individual differences in demographic, trait or context variables, and genetic markers will be investigated as predictors of vulnerable or resilient trajectories by other authors in a priori planned papers. Thus, these additional analyses will be part of future publications of the CovSocial project. We have extended this future outlook in the discussion section (p. 23). To avoid duplication of analyses and to keep the current paper focused, we thus prefer to refrain from additional analyses that include only few selected covariates. We hope for your understanding.
4. Study design- one of the limitations of this study is the fact that it has a retrospective longitudinal design - the perception may have been altered with the experience of the ansiogenic situation (world pandemic), since the data were collected between September 11th, 2020 to December 7th, 2020, should soon always approach as perception of the individual.
We have adapted our language throughout the discussion section to emphasize that all data interpretations and conclusions are based on retrospective subjective assessments. We now particularly refer to the measured phenomena as subjective perceptions.
5. Measures- the concepts listed in the literature were evaluated and supported in validated instruments. The use of only some items of the scales, analysed through the means may have led to some biases. Important to explore the issue.
We apologize that the description of our measures and analyses must have caused a misunderstanding. We used full validated scales, not only some items out of those scales. The items of the validated scales were summed or averaged according to the scale manuals. Our self-generated items were first tested for internal consistencies for each of the construct that we were aiming to measure, and composites were only built by a mean across those items when Cronbach’s Alpha was ≥ .70. Supplement 2 shows internal consistencies of scales and pools of items. We clarified this procedure in the Data Analysis and Results sections section (p.10 & p.13), and additionally refer to Supplement 2 and Supplement 1 Appendix B for information on item and scale statistics.
6. Data Analysis- A significance level of α = .05 was used for all analyses. Orp?
The significance level is denoted by α. It is the defined probability of the study to reject the null hypothesis when it is true, and thus it is the threshold for p.
7. Conclusions- Repetitive and very extensive text that should be, in my point, divided into two or even three articles:
The discussion section is now divided into four subsections: 1) factor structure, 2) within construct changes, 3) between-construct correlations, and 4) limitations.
-Factor analyses - identifying and validating the most probable factor structure for the three main psychological constructs that were measured based on a broad variety of different indicators, i.e., vulnerability, resilience and social cohesion. Very interesting.
-Retrospective longitudinal design - identifying how these latent factors systematically change over time during the COVID-19 pandemic in Berlin (January 2020; March/April 2020 and June 2020. Although they use strategies to activate memories, we are talking about a very intense period in the psychoemotional organization, which brings high risks of bias. They should address/further explore this issue.
In the limitations section, we now emphasize that the memory-bias of retrospective data assessment may be particularly pronounced due to the ongoing pandemic.
-Investigating individual and at identifying the correlative interrelationships between these constructs and their changes over time. These features are not explored in detail. It would be important to have more sample data and carry out their exploratory analyses.
We thank the reviewer for this comment and agree that additional exploratory analyses would enrich an understanding of the lockdown-related impacts. In the revised manuscript, we now highlight gaps of current findings and open questions for future studies (p 23, 25). Please also see response 3 for our general approach and Supplement 1 for further information on the CovSocial project.